# Development and Validation of a Scale for Christian Character Assessment for Young Children[1]

**Sungwon Kim**

Industrial Education Department, Chongshin University, 143 Sadang-ro, Dongjak-gu, Seoul 06988, Korea; swkim@chongshin.ac.kr

**Abstract:** The emphasis on character education that has emerged from the Nuri national curriculum and the Character Education Act is leading the direction of Korean education. However, the lack of a proper scale of character assessment—especially Christian character for young children—has caused uncertainty in related studies. This study aims to develop and validate a scale that assesses Christian character for young children. The data was collected from 257 (study 1) and 405 (study 2) Christian children who attend church and kindergarten or day care center. Within 12 factors, 67 questions were developed, which were subsequently refined to 39 questions by seven professors. The Christian character scale for young children was finalized to twenty-four questions through exploratory factor analysis and confirmatory factor analysis within four factors: *piety*, *self-control/harmony*, *responsibility/independence*, *and caring/respect*.

**Keywords:** Christian character; young children; scale development; validation

## 1. Introduction

Due to changes in family structures, such as low birth rates, an increase in the number of dual-income families, and the emergence of various family types, opportunities for character education at home have diminished (Cha and Na 2016; Kim et al. 2015). Moreover, young children are immersed in TV, smart phones, and computer games, losing the ability to plan and enjoy their playtime by themselves, and are gradually losing opportunities for natural character formation with their friends through play (Choi and Yousun 2015). In modern society, in which the self-centered approach is prevalent, young children have come to learn selfishness instead of caring and respect of others (Cha and Na 2016), making it difficult for children to form character and happiness amid the feverish developments in childhood education (Choi and Yousun 2015). Furthermore, exposure to adverse life experiences—maltreatment and violence, loss events, intra-familial problems, school and interpersonal problems—during early life can eventuate physical illness, as well as the developmental of psychopathological disorders including depression, anxiety, antisocial behavior, or suicidal behavior in later development (Serafini et al. 2015).

Along with these current factors, the history of education shows the distortion of basic concepts of early childhood education in Korea. Education emphasizes uniform teaching methods and textbook-based education since the introduction in the 1970s of Piaget's theories of cognitive development. Consequently, because early childhood education has been closely identified with very early education, specialized education, and intellectual education, young children have come to be

---

nurtured with an imbalance between intellectual development on the one hand and aspects of social, emotional and moral development on the other (Roh 2010).

[1] This work was supported by the Ministry of Education of the Republic of Korea and the National Research Foundation of Korea (NRF-2017S1A5A8022121).

Reflecting on the outcomes of such education and searching for a remedy to the various social ills and crimes have led to character education as a solution. For example, the first directive for a national curriculum for young children—Nuri Curriculum—is focused on basic life habits and the development of good character such as order, caring, and cooperation. Subsequently, after recognizing the limitations of recent cognitive-centered education, Korea enacted the Character Education Promotion Act in 2015 (Park 2016).

In order to provide proper education in a particular area, it is essential to precisely define the components of education and develop measurement scales for such areas. However, character scales for young children were used after revising scales for elementary students (Kim 2014; Sohn and Kim 2016) or translating foreign measurement scales (Kim 2014). Character scales for young children have begun to be developed only recently (Na and Kim 2014; Baek and Lim 2015; Seo et al. 2017).

Inadequacy of the factors and questions from the already existing scales was found. Cha and Na (2016) used a character scale composed of the *formation of basic living habits*, *self-concept*, *community spirit*, *physical development*, and *social development* based on Conners' character scale. The composition of these factors, and the disharmony between factors and questions can be problematic—such as the *physical development* factor's inclusion of the questions, "he/she cannot control his/her emotions well" and "he/she cries easily upon minor rebuke or language." The representative Christian character scale was developed by Good Tree Character School (2007) and consists of 12 factors and 72 questions covering empathy and conscience. The questions in the scale are general character questions excluding Christian elements such as "My child tends to carefully watch his/her surroundings and focus well," "My child knows he/she has a lot of great strengths," and "My child can express his/her feelings well." Furthermore, certain questions in the *joy* factor such as "My child treats others as precious and does not speak ill of them," "My child observes the rules and order of the community he/she belongs to," and "My child leads an orderly life with good eating habits" are not consistent with the factor name. The factors of young children's character chosen by recent research is presented in Table 1.

**Table 1.** Factors of Young Children's Character.

| | | |
|---|---|---|
| Ministry of Education, Science and Technology (2011) | | order, respect elders and honor parents, caring, respect, cooperation, conflict resolution |
| Choi and Lim (2013) | basic living habits | courtesy, orderliness, cleanliness, dietary habits, well organized |
| | social emotion | self-concept, stability, self-control, respect, cooperation, attentive listener, conciliatory, caring, kindness, affection, trustworthiness, communication, gratefulness, generosity, community spirit |
| | ethics and morality | public orderliness, diligence, courage, positive contribution (positivity), honesty, love, responsibility, trust, patience, devotion to parents, fairness, obedience, harmony, respect for other cultures, respect for traditional ethics, respect for life, patriotism |
| Kim (2014) | | trust, respect, responsibility, fairness, caring, civic spirit |
| Na and Kim (2014) | interpersonal values | sharing, caring, courtesy, patience, respect, orderliness, responsibility, cooperation, honesty |
| | individual values | identity formation, positive attitude, persistence, self-discipline, diligence |
| | social values | world citizen spirit, community service, commitment as a member of society |
| Baek and Lim (2015) | | sharing, caring, courtesy, patience, respect, order, responsibility, cooperation, honesty |
| Seo et al. (2017) | | interested in problem solving, finding alternatives, initiating action |
| Lee (2012) * | empathy | attentive listening, positive attitude, happiness, caring, gratefulness, obedience |
| | conscience | patience, responsibility, self-control, creativity, honesty, wisdom |
| Jeoung et al. (2014) * | | piety (holiness), love (compassion and mercy), self-control, patience, caring (gentleness, attentive listening, kindness), cooperation, peace (harmony), responsibility (loyalty, trust), honesty (sincerity, truthfulness), benevolence (goodness, generosity), joy, respect |

\* Christian character factor.

It is notable that most of character theories and scales include *caring* as a factor and that there are scales which include *basic living habits* and *civil spirit* as factors.

Early researchers were asked to develop Christian character scales (Jeoung et al. 2013; Kim 2016; Park 2012; Roh 2010). For example, Park (2012) was unable to conduct research with young children regarding Christian sharing due to the absence of scales. She suggested a need to develop an evaluation scale for teachers to measure varieties of Christian character. Due to the limitations of early research and requests for the development of a Christian character scale by the advanced researcher, this research aims to develop a teacher-rated Christian character scale for 4- and 5-year-olds and ensure validity. Based on these research objectives, the research questions are:

1. What are the factors of a Christian character scale for young children?
2. What is the validity of the Christian character scale for young children?
3. What is the reliability of the Christian character scale for young children?

## 2. Methods

### 2.1. Research Subjects

The data set of 762 participants was divided into two samples from which exploratory factor analysis (EFA) (Sample 1, *N* = 257) and confirmatory factor analysis (CFA) (Sample 2, *N* = 405) were conducted. When employing a CFA with the same data from an EFA, the results are not informative, and can be inaccurate. Therefore, analyzing a secondary CFA with new data is recommended (Henson and Roberts 2006). Following this principle, this study used separate samples for the EFA and CFA. The participants were 4- and 5-year-olds who attend church and kindergarten or day care center and their character level corresponding to each question was rated by their kindergarten/day care center teachers using a 5-point Likert scale. For Samples 1 and 2, the average age is 61.50 months and 62.62 months, and consists of 133 boys (51.75%) and 124 girls (48.25%), and 203 boys (50.1%) and 202 girls (49.9%), respectively.

### 2.2. Research Procedure

#### 2.2.1. Factor Development

Christian character is fundamentally different from the concept of character contemplated under general psychology and education studies. Character based on humanism is focused on oneself, whereas the focus of Christian character is different because it resembles God's character (Jeoung et al. 2013; Park 2012; as cited in Kim and Shin 2018). However, because character research is rarely conducted on a theological basis, it could lead to moralism, or the development of sociability and creativity for success (Kim 2012). In this study, Christian character is defined as "building relationships with God, others, and self, based on God's Word."

The character virtues selected are from those discussed in the Nuri Curriculum (Ministry of Education, Science and Technology 2012), the Character Education Promotion Act, the earlier-developed Christian character programs called Bitu–Baro character school, the Good Tree Character School, and Saemmul Kindergarten. Those virtues were presented in Jeoung et al. (2013)'s research, which analyzed Christian character virtues appropriate for young children, and Baek and Lim (2015)'s research, which developed a character scale for young children. Character virtues from the previous studies are listed in Table 2 (as cited in Kim and Shin 2018).

**Table 2.** Character Virtues for Young Children.

| Developer / Virtues | National Law Curriculum | | Christian Character Program | | | Christian Character Constucts | Character Scale |
|---|---|---|---|---|---|---|---|
| | Nuri Curriculum (2012) | Character Education Promotion Act (2015) | Bitu–Baro Character School | Saemmul Kindergarten | Good Tree Character School | Jeoung et al. (2013) | Baek and Lim (2015) |
| gratitude | | | • | • | • | | |
| humility | | | | • | | | |
| piety | | | | | | • | |
| attentive listening | | | • | • | • | | |
| honor | | | | • | | | |
| positive attitude | | | | | • | | |
| joy | | | | | • | • | |
| sharing | • | | • | | | | • |
| caring | • | • | • | | • | • | • |
| love | | | • | • | | • | |
| communication | | • | | | | | |
| obedience | | | • | • | • | | |
| goodness | | | | | | • | |
| courtesy | | • | | | | | • |
| courage | | | • | | | | |
| forgiveness | | | • | • | | | |
| patience | | | • | | • | • | • |
| mercy | | | | | | | |
| honesty | | • | • | | • | • | • |
| self-control | | | | • | • | • | |
| prudence | | | | • | • | | |
| respect | • | • | | | | • | • |
| wisdom | | | • | | | | |
| honest | | | | • | | | |
| orderliness | • | | | | | | • |
| responsibility | | • | | | • | • | • |
| creativity | | | | | • | | |
| peace | | | | | | • | |
| cooperation | • | • | • | | | • | • |
| honoring parents | • | | | | | | |

The process of reducing the factors and questions was conducted with two validity tests. Twenty-one experts were asked to evaluate 30 character virtues that were discussed at least once in the above material, using a 5-point Likert scale after categorizing them into "appropriateness as Christian character," "developmental appropriateness," and "feasibility of measurement." The participants were seven professors from fields of study relevant to Christian character development of young children and fourteen experts with master's degrees or higher education and more than 5 years of work experience with young children. From the first validity test, 12 virtues—*piety*, *love/forgiveness*, *attentive listening*, *obedience*, *caring/sharing/goodness*, *respect/manners*, *gratefulness/joy*, *peace/cooperation*, *order*, *responsibility*, *honesty*, and *self-control/patience*—were extracted. The results showed some factors overlapped with those appearing 3 times in the earlier studies as shown in Table 2. The character virtues that were selected three or more times were: *gratefulness*, *attentive listening*, *sharing*, *caring*, *love*, *obedience*, *patience*, *self-control*, *honesty*, *respect*, *responsibility*, and *cooperation*. Except for *respect* and *piety*, these are identical with the above-mentioned experts' opinions.

### 2.2.2. Item Development and Content Validity

As a next step in the development of the scale, 67 questions covering the 12 virtues were devised. The questions were evaluated by six professors with majors in early childhood education, theology and Christian education and one expert who completed a doctoral degree in early childhood education and has work experience of more than 30 years in education. Pursuant to the above process, 12 virtues and 39 questions were finally selected.

### 2.2.3. Pilot Study

It is recommended that twenty-five or more samples for a pilot study are needed (Patten 2001). A pilot test was conducted with 30 children—5-year-olds in R Kindergarten, located in Seoul. A teacher that participated in the pilot test majored in Early Childhood Education in a 4-year university course and worked for 2 years and 7 months as a teacher. The pilot test revealed two questions which were

unclear or difficult to rate, these were revised as follows: No. 13, "A child knows that God loves him/her and sent Jesus for himself/herself." was modified to "A child knows that Jesus loves him/her." No. 31, "A child knows that everything he owns comes from God and is grateful for it." was revised to "A child is satisfied and grateful for resources given by God."

## 3. Results

### 3.1. Data Screening

Normality tests using skewness and kurtosis scores were examined. Skewness ranged from −0.24 to −0.00 and kurtosis ranged from −0.94 to 2.14 for Sample 1 and skewness ranged from −0.65 to −0.20 and kurtosis ranged from −0.89 to 0.46 for Sample 2. Different standards regarding normality are suggested by different statisticians: values between −0 and +2 for skewness and kurtosis are recommended (Field 2009), or less than 3 for kurtosis are permitted (Bae 2017; Lee 2017). Therefore, normal distribution of data was not violated.

### 3.2. Exploratory Factor Analysis

Exploratory factor analysis was performed using a principal component analysis running with Varimax rotation. Prior to the extraction of the factors, the Kaiser–Meyer–Olkin (KMO) Measure of Sampling and Bartlett's Test of Sphericity were used to assess the suitability of the respondent data for factor analysis. In this study, the KMO value (0.933) and Bartlett's test value ($\chi^2$ (276) = 3830.05, $p$ = 0.000) indicate that factor analysis is appropriate (Cho 2016a). Items with factor loadings below 0.4 or items of cross-loadings were removed from the analysis.

The four factors that had eigenvalues greater than one were extracted, and the 63.99% variance was explained. The first factor, entitled *Piety*, included seven items, representing 20.23%. The second factor, entitled *Self-control/Harmony,* included eight items, representing 19.40%. The third factor, entitled *Responsibility/Independence*, included five items, representing 14.54%. The fourth factor entitled *Caring/Respect*, included four items, representing 9.82%. Rotated factor loadings, variances, and eigenvalues are provided in Table 3.

**Table 3.** Factor Loadings.

| Items (A Child) | Factor 1 | Factor 2 | Factor 3 | Factor 4 | Communality |
|---|---|---|---|---|---|
| Factor 1: Piety | | | | | |
| 37. is happy to be attending church | 0.820 | 0.162 | 0.102 | 0.136 | 0.727 |
| 13. knows that Jesus loves him/her | 0.816 | 0.213 | 0.246 | −0.015 | 0.773 |
| 25. expresses thoughts about church and faith | 0.803 | 0.107 | 0.032 | 0.208 | 0.700 |
| 16. tries to listen and obey God's words | 0.759 | 0.228 | 0.256 | 0.194 | 0.731 |
| 31. is satisfied and grateful for resources given by God | 0.757 | 0.280 | 0.102 | 0.188 | 0.698 |
| 34. treasures what God has created and cares for it | 0.744 | 0.379 | 0.139 | 0.127 | 0.732 |
| 1. likes God's words, praise, prayer, and worship | 0.642 | 0.056 | 0.339 | 0.184 | 0.563 |
| Factor 2: Self-control/Harmony | | | | | |
| 36. can wait even if his/her needs are not immediately met | 0.216 | 0.799 | 0.203 | 0.130 | 0.742 |
| 28. throws a tantrum in order to achieve a difficult request to be accepted | 0.129 | 0.742 | 0.308 | 0.123 | 0.677 |
| 29. is conciliatory | 0.219 | 0.687 | 0.214 | 0.346 | 0.685 |
| 32. adjusts to and cooperates with his/her friends well in order to accomplish a collective task | 0.302 | 0.678 | 0.187 | 0.253 | 0.650 |
| 35. maintains values, convictions, and good behavior regardless of changing circumstances | 0.239 | 0.678 | 0.171 | 0.070 | 0.551 |
| 33. knows how to take turns and wait for his/her turn | 0.190 | 0.658 | 0.383 | 0.148 | 0.638 |
| 24. does not complain or have a temper tantrum when he/she is unable to get what he/she wants | 0.102 | 0.636 | 0.338 | 0.078 | 0.536 |
| 14. forgives his/her friends' faults | 0.300 | 0.415 | 0.324 | 0.179 | 0.400 |

| Items (A Child) | Factor 1 | Factor 2 | Factor 3 | Factor 4 | Communality |
|---|---|---|---|---|---|
| Factor 3: Responsibility/Independence | | | | | |
| 21. keeps track of time (play, meals, arrival times) | 0.158 | 0.247 | 0.769 | 0.115 | 0.691 |
| 11. selects what is right and good | 0.230 | 0.329 | 0.684 | 0.095 | 0.638 |
| 10. completes a task given to him/her | 0.138 | 0.349 | 0.674 | 0.151 | 0.618 |
| 22. admits his/her faults and takes responsibility | 0.206 | 0.346 | 0.631 | 0.230 | 0.613 |
| 18. uses refined and honorific language to adults | 0.175 | 0.272 | 0.630 | 0.278 | 0.579 |
| Factor 4: Caring/Respect | | | | | |
| 6. says hello to adults first with a courteous manner | 0.149 | 0.099 | 0.341 | 0.721 | 0.668 |
| 7. expresses his/her gratefulness to a person who provides assistance | 0.283 | 0.023 | 0.400 | 0.686 | 0.712 |
| 5. gets along well with special needs individuals (foreigner, disabled, lonely child) | 0.099 | 0.359 | 0.005 | 0.656 | 0.569 |
| 2. reacts to other's difficulties sensitively and helps out | 0.281 | 0.285 | 0.102 | 0.543 | 0.466 |
| % of Variance | 20.23 | 19.40 | 14.54 | 9.82 | |
| Eigenvalues | 10.50 | 2.34 | 1.40 | 1.12 | |

## 3.3. Confirmatory Factor Analysis

### 3.3.1. Model Fit

An initial confirmatory factor analysis was conducted to assess 39 items, within seven factors, based on exploratory factor analysis results. However, the original model fit failed to meet the standards ($\chi^2(681) = 2279.50$, $p = 0.000$; standardized root mean square residual (SRMR) = 0.062; goodness-of-fit-index (GFI) = 0.761; comparative fit index (CFI) = 0.843; Turker-Lewis index (TLI) = 0.830; root mean square error of approximation (RMSEA) = 0.076 (low 0.073; high 0.080). The revised model with five factors and 26 items provided a highly satisfactory fit to the data ($\chi^2(289) = 825.82$, $p = 0.000$; SRMR = 0.048; GFI = 0.868, CFI = 0.914; TLI = 0.903; RMSEA = 0.068 (low 0.062; high 0.073)). The problems of this model are that some of the values do not match the criteria and factor five includes only two items (no. 8 and no. 14), whereas statisticians recommend at least three items per factor (Yu 2012).

Based on the results of exploratory factor analysis that supports confirmatory factor analysis with the proper factors and item–factor relationship (Cho 2016a), the modified model was comprised of 24 items within four factors. The modification indices revealed that covariances of e9 and e10, and e4 and e7, are 76.56 and 33.33, respectively. Similarity in meaning was found between question no. 24 (e9) and no. 28 (e10), in that both ask about the child's strong will. There was also a similarity between question no. 25 (e4) and no. 37 (e7), in that these have to do with a child's faith. This researcher, therefore, covaried those error terms and the model fit was slightly improved.

With these changes, the model produced highly satisfactory results ($\chi^2(244) = 596.41$, $p = 0.000$; SRMR = 0.046; GFI = 0.892, CFI = 0.938; TLI = 0.930; RMSEA = 0.060 (low 0.054; high 0.066)). Table 4 presents the model fit indices for the original and revised models.

**Table 4.** Goodness-Of-Fit Indices from Confirmatory Factor Analysis.

| | $\chi^2$ | df | SRMR | CFI | TLI | RMSEA | |
|---|---|---|---|---|---|---|---|
| | | | | | | Low | High |
| 39 items | 2279.50 | 681 | 0.062 | 0.843 | 0.830 | 0.073 | 0.080 |
| 26 items | 825.82 | 289 | 0.048 | 0.914 | 0.903 | 0.062 | 0.073 |
| 24 items | 596.41 | 244 | 0.046 | 0.938 | 0.930 | 0.054 | 0.066 |
| Cutoffs | | | ≤0.08 | ≥0.90 | ≥0.90 | ≤0.06 | |

With the revised model, the standard coefficients of each factor appeared as: factor 1: 0.68−0.85, factor 2: 0.62−0.78, factor 3: 0.67−0.76, and factor 4: 0.59−0.80. These standard coefficients and graphic illustrations of the four-factor model with 24 items is presented in Figure 1.

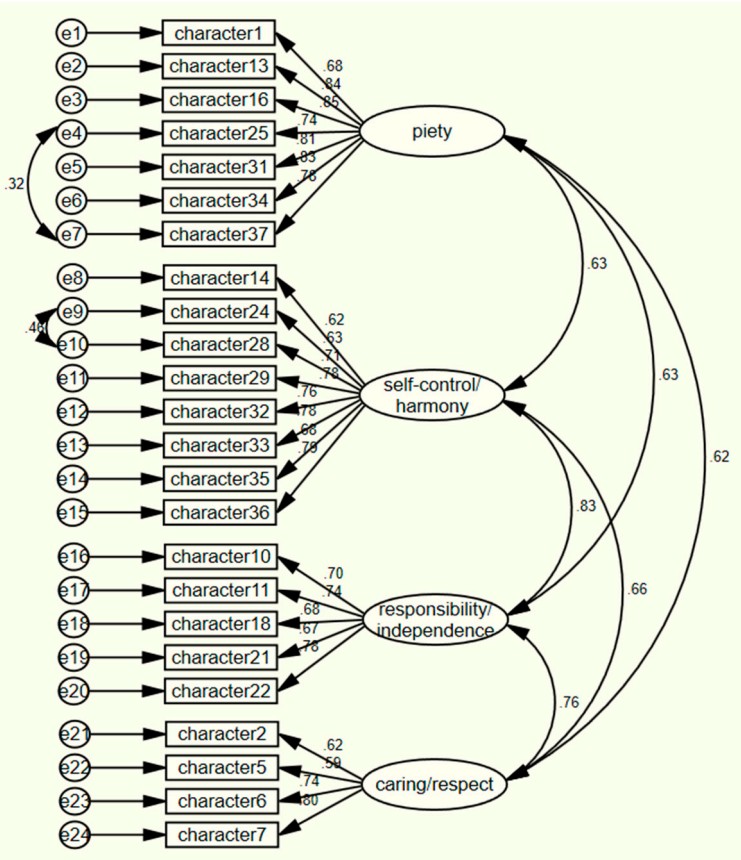

**Figure 1.** The path diagram of the final four-factor model.

### 3.3.2. Convergent and Discriminant Validity

Confirmatory factor analysis was used to assess the measurement model with respect to convergent and discriminant validity (Yu 2012). Large standardized factor loadings, which indicated large inter-correlations among items associated with the same latent variable, support convergent validity (Brown 2015). The averaged variance extracted (AVE) and construct reliability (CR) scores are higher than the cutoffs of 0.5 and 0.7, respectively (Cho 2016a; Yu 2012) (see Table 5).

**Table 5.** Convergent Validity.

| Construct | Item | Standardized Regression Weight | t-Value (CR) | CR (Construct Reliability) | AVE (Averaged Variance Extracted) | Cronbach' α |
|---|---|---|---|---|---|---|
| Piety | 1 | 0.683 | 14.366 | 0.945 | 0.712 | 0.919 |
| | 13 | 0.836 | 18.391 | | | |
| | 16 | 0.851 | 18.811 | | | |
| | 25 | 0.738 | 19.158 | | | |
| | 31 | 0.811 | 17.687 | | | |
| | 34 | 0.831 | 18.246 | | | |
| | 37 | 0.780 | Fix | | | |
| Self-control/Harmony | 14 | 0.618 | 12.744 | 0.936 | 0.650 | 0.896 |
| | 24 | 0.627 | 12.948 | | | |
| | 28 | 0.707 | 14.934 | | | |
| | 29 | 0.778 | 16.804 | | | |
| | 32 | 0.762 | 16.367 | | | |
| | 33 | 0.780 | 16.866 | | | |
| | 35 | 0.681 | 14.275 | | | |
| | 36 | 0.789 | Fix | | | |
| Responsibility/Independence | 10 | 0.700 | 14.240 | 0.906 | 0.659 | 0.837 |
| | 11 | 0.739 | 15.161 | | | |
| | 18 | 0.679 | 13.754 | | | |
| | 21 | 0.674 | 13.642 | | | |
| | 22 | 0.781 | Fix | | | |
| Caring/Respect | 2 | 0.620 | 11.967 | 0.862 | 0.614 | 0.779 |
| | 5 | 0.593 | 11.399 | | | |
| | 6 | 0.735 | 14.285 | | | |
| | 7 | 0.801 | Fix | | | |

Model fit $\chi^2(244) = 596.409$, $p = 0.000$; SRMR = 0.046, CFI = 0.938;
TLI = 0.930; RMSEA = 0.060 (low 0.054, high 0.066)

In regards to discriminant validity, correlations are less than 0.723 and the squared correlations are lower than AVE (Yu 2012). The correlations between factors and AVE presented in Table 6 prove discriminant validity.

**Table 6.** Inter-Correlations between Factors.

| Factors | Factor 1 | Factor 2 | Factor 3 | Factor 4 | *M* | *SD* |
|---|---|---|---|---|---|---|
| Factor 1 | [0.712] | | | | 3.87 | 0.67 |
| Factor 2 | 0.634 ** | [0.650] | | | 3.89 | 0.58 |
| Factor 3 | 0.537 ** | 0.723 ** | [0.659] | | 4.03 | 0.57 |
| Factor 4 | 0.532 ** | 0.581 ** | 0.622 ** | [0.614] | 3.89 | 0.59 |

** $p < 0.01$, [ ] AVE.

## 4. Discussion and Conclusions

From this research, pursuant to the two validity tests, the pilot test, main test, and statistical analysis, 24 questions within four factors were selected for the Christian character scale for young children. These four factors are *piety/spirituality*, *self-control/harmony*, *responsibility/independence*, and *caring/respect*. In this study, a high score means well-developed biblically-sound relationships with God, others, and self.

By examining the excluded questions, it was found that Christian character requires a higher-level active aspect covering piety, self-control, responsibility, and caring among others—going beyond universal character attributes such as attentive listening/sympathy, good relationship maintenance, offering help, or resolution of difficulties. Looking at questions in more detail, no. 24 and no. 28 not complaining and pestering when their wishes are not fulfilled and no. 33 and no. 36 not waiting while keeping turns, and with patience in the *self-control* factor, are difficult to implement because of the child's need to go against their will, but nevertheless, their need to be trained. According to

Yates and Yates (1992), individual happiness comes from self-control—one of the Christ-like virtues. Overeating, overcommitting, wasteful spending habits, intemperance of the tongue, weak will, or being a workaholic resulting from a lack of self-discipline might lead to trouble.

No. 35 and no. 39, not changing words and actions in a way advantageous to themselves and keeping promises, are strictly necessary virtues that meet the demands of the times and the countries in which serious ethical and moral issues are becoming social issues. Furthermore, no. 10, not completing his/her tasks till the end, and no. 22, not acknowledging his/her faults and taking responsibilities, which are covered by the *responsibility* factor, are important for the same reasons. No. 11 appears to be a desirable question that needs to be further emphasized in Christian character because this researcher believes that conveying the Christian and social values of goodness and righteousness is one of the most important missions of Christian educators. As the people of God "will be called oaks of righteousness . . . I, the LORD, love justice" (Is 61:3, 8), the substance and heart of the active Christian life is righteousness. Christians are forgiven and reconciled by God's grace and mercy and then made righteous, so that righteousness would be expressed through Christians' lives. The fourth Beatitude conveys the idea that persons of good character desire righteousness to be integrated into their daily lives (Gill 2000).

Caring has been categorized as a factor, which is a consistent outcome of the fact that it was presented as a factor in earlier character-related research (Baek and Lim 2015; Jeoung et al. 2013). It is the character virtue most curricula and scholars chose for young children, as shown in Table 2 and recent dissertations (Oh 2016; Park 2019) and research papers (Cho 2016b; Jahng and Song 2016; Lim et al. 2019) on young children that have been published focusing only on caring. Caring is the most basic and important virtue, in that it might be expanded into respect, cooperation, sharing, order, and filial duty (Oh 2016). The fact that caring is a separate factor in this scale proves the assumption that caring is a foundational virtue of character in general education, as well as Christian character.

A surprising fact is that for questions inquiring about the relationship with God, none was excluded and all of them were bound as a single factor. The relationship with God is the basis of a Christian's search for the good. Christian ethics of good character should not be separated from theology. Unless God is at the center, behaving ethically is not possible (Gill 2000). As Gill (2000), Jeoung et al. (2014), and Smith (2016) argued, the relationship with God is the foundation of the Christian's good character, re-verifying this point in this research is a meaningful finding. A correct relationship with oneself and others is the direct result of the relationship with God, the Creator (Kim 2013). In this sense, piety indicates the relationship with God, and self-control/harmony and responsibility/independence represent the relationship with self, and caring/respect reflects the relationship with others—all factors of this study cover the realm of relationships.

The significance of this research is that it has developed a Christian character scale for young children, which is appropriate for young children's development, is theologically sound, and has undergone all the statistical procedures necessary for such a scale development. Secondly, as explained above, although there have been many studies that, after developing and implementing Christian character programs, have failed to verify any effects on character due to the lack of a scale, now, based on this research, studies of young Christian children's characters will be conducted with greater accuracy and more widely. Lastly, since systematic and differentiated approaches to Christian character education are required (Kim 2012), it is expected that this study could establish a milestone for what to teach at church, home, and day schools. All children need to learn the virtues of positive character development but, in most cases, this is not a reality. Teachers at every stage, therefore, should know which character virtues need to be learned and how to teach these to children (Stronks and Stronks 2008). In this sense, the findings of this study might be informative regarding character virtues and content for teachers and leaders who are responsible for the Christian character education of children.

Even though this manuscript is significant, it also has limitations. Data was collected from early-childhood teacher evaluations of young children. There are limits to generalizing the results of

the study for rating by Sunday school teachers or parents. In these cases, the use of this scale might be possible after further factor analysis and reliability testing.

It is expected that character education for young children, which is required in society, and a justifiably essential content of Christian education, will gain more attention, be implemented more effectively, and its outcomes will be measured more scientifically and reasonably.

**Funding:** This research was funded by the Ministry of Education of the Republic of Korea and the National Research Foundation of Korea, NRF-2017S1A5A8022121.

**Conflicts of Interest:** The author declares no conflict of interest.

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
