# Peer review of "Development and Validation of a Scale for Christian Character Assessment for Young Children1"

_religions, doi:10.3390/rel10050318_

Round 1

Reviewer 1 Report

This is, in summary, an interesting paper aimed to develop and validate a scale that assesses the Christian character for young children based on the data collected from 257 (study 1) and 405 (study 2) Christian children. The authors reported that within 12 factors, overall 67 questions were developed, which were subsequently refined to 39 questions. The authors added that the Christian character scale for young children was finalized to 24 questions through exploratory factor analysis and confirmatory factor analysis within four factors: piety, self control/harmony, responsibility/independence, and caring/respect.

The authors may find as follows my main comments/suggestions.

First, when within the Introduction section, the authors correctly stated that young children are immersed in TV, smart phones, and computer games—losing the ability to plan and enjoy their playtime by themselves, and are gradually losing opportunities for natural character formation with their friends through play, the link between adverse life events (maltreatment and violence, loss events, intra-familial problems, school and interpersonal problems) and negative outcomes such as suicidal behavior in young adolescents might be also reported. Based on the results of a systematic review, the number of the experienced adversities or negative life events in youths and adolescents seemed to have a positive dose-response relation with youth suicidality. In order to briefly focus on the specified association (although i understand that the link between adverse life events, and negative outcomes in young people is not the main topic of this paper), i suggest to cite within the main text the paper published on Eur Child Adolesc Psychiatry in 2015 (PMID: 26303813).

In addition, as the authors reported extensively the most important aims/objectives of this paper, the main study hypotheses should be similarly described in a more detailed manner.

Moreover, the rationale through which the total sample was divided into two subsamples from which EFA (Sample 1, N=257) and CFA (Sample 2, N=405) were conducted, needs to be better described for the general readership.

Furthermore, the most relevant implications related to the main findings (the 4 factors piety/spirituality, self-control/harmony, responsibility/independence, and caring/respect obtained using  exploratory factor analysis and confirmatory factor analysis) need to be reported immediately.

Importantly, the most relevant shortcomings/limitations of this manuscript should be cited and discussed in a more detailed manner as their description is really poor and limited based on the current version of the present manuscript.

Finally, what is the take-home message of this paper? While the authors stated that character education for young children is expected to gain more attention, and be implemented more effectively, they failed, in my opinion, to report the most relevant conclusive remarks about the proposed conclusion. Specifically, how character education for young children might gain more attention? Which are the suggested strategies in order to improve more effectively character education for young children? Here, more details/information are needed.

Author Response

I applied all the suggestions from Reviewer 1 except the following:

In addition, as the authors reported extensively the most important aims/objectives of this paper, the main study hypotheses should be similarly described in a more detailed manner.

I could not find hypotheses in scale-development papers—only purpose and problem statements. If you teach me how to write hypotheses, I will add them.

Reviewer 2 Report

This is useful study addressing an issue of current concern is S. Korea,  mandatory Christian education.  The authors provide a reasonable constructed and reliable measure of character in light of a Christian. biblical model.  The EFA and CFA are appropriate and the scale should be reliable in assessing character formation.  The comparison to previous scales, largely secular or humanistic is useful.  However, one issue that ought to be mentioned and is relevant for future research is that rating of all children was done across all items.  Hence, there is no empirical basis for concluding that piety (major factor) positively relate to other factors insofar as once raters have decided on issue of piety it likely colors how they rate other factors.  The uniqueness of piety is better established by independent raters judging children on each factor separately to see if their is truly a unique contribution to Christian (biblical) character separate from the other already established more humanistic or secular characteristics.

Author Response

However, one issue that ought to be mentioned and is relevant for future research is that rating of all children was done across all items.  Hence, there is no empirical basis for concluding that piety (major factor) positively relate to other factors insofar as once raters have decided on issue of piety it likely colors how they rate other factors.  The uniqueness of piety is better established by independent raters judging children on each factor separately to see if their is truly a unique contribution to Christian (biblical) character separate from the other already established more humanistic or secular characteristics.

The original 67 items within 12 factors were initially developed and items were reduced to 39 through a content validity test narrowed by a panel of experts. During this process, the 8 questions in the piety category were reduced to 4, and these 4 were used in the main survey. The additional 3 items comprising the final 7 piety factors were from other categories in which items were integrated with a spiritual meaning (e.g. 16. tries to listen and obey God’s words.-> obedience; 31. is satisfied and grateful for resources given by God.->gratefulness/joy; 34. treasures what God has created and cares for it.-> responsibility). The relationship between piety and other factors was proven from statistical data with correlation coefficient in Table 6(piety & self control/harmony r= .634; piety & responsibility/ independence r= .537; piety & caring/respect r= .532).

Round 2

Reviewer 1 Report

In the revised paper, the authors successfully addressed most of the major questions raised by Reviewers improving both the main structure and quality of the main text. I have no further additional comments.